# Deep Learning in Physiological Signal Data: A Survey

**DOI:** 10.3390/s20040969

**Published:** 2020-02-11

**Authors:** Beanbonyka Rim, Nak-Jun Sung, Sedong Min, Min Hong

**Affiliations:** 1Department of Computer Science, Soonchunhyang University, Asan 31538, Korea; 2Department of Medical IT Engineering, Soonchunhyang University, Asan 31538, Korea; 3Department of Computer Software Engineering, Soonchunhyang University, Asan 31538, Korea

**Keywords:** deep-learning, machine learning, physiological signals, 1D signal data analysis

## Abstract

Deep Learning (DL), a successful promising approach for discriminative and generative tasks, has recently proved its high potential in 2D medical imaging analysis; however, physiological data in the form of 1D signals have yet to be beneficially exploited from this novel approach to fulfil the desired medical tasks. Therefore, in this paper we survey the latest scientific research on deep learning in physiological signal data such as electromyogram (EMG), electrocardiogram (ECG), electroencephalogram (EEG), and electrooculogram (EOG). We found 147 papers published between January 2018 and October 2019 inclusive from various journals and publishers. The objective of this paper is to conduct a detailed study to comprehend, categorize, and compare the key parameters of the deep-learning approaches that have been used in physiological signal analysis for various medical applications. The key parameters of deep-learning approach that we review are the input data type, deep-learning task, deep-learning model, training architecture, and dataset sources. Those are the main key parameters that affect system performance. We taxonomize the research works using deep-learning method in physiological signal analysis based on: (1) physiological signal data perspective, such as data modality and medical application; and (2) deep-learning concept perspective such as training architecture and dataset sources.

## 1. Introduction

Deep Learning has succeeded over traditional machine learning in the field of medical imaging analysis, due to its unique ability to learn features from raw data [1]. Objects of interest in medical imaging such as lesions, organs, and tumors are very complex, and much time and effort is required to extract features using traditional machine learning, which is accomplished manually. Thus, deep learning in medical imaging replaces hand-crafted feature extraction by learning from raw input data, feeding into several hidden layers, and finally outputting the result from a huge number of parameters in an end-to-end learning manner [2]. Therefore, many research works have benefited from this novel approach to apply physiological data to fulfil medical tasks.

Physiological signal data in the form of 1D signals are time-domain data, in which sample data points are recorded over a period of time [3]. These signals change continuously and indicate the health of a human body. Physiological signal data categories fall into characteristics such as electromyogram (EMG), which is data regarding changes to skeleton muscles, electrocardiogram (ECG), which is data regarding changes to heart beat or rhythm, electroencephalogram (EEG), which is data regarding changes to the brain measured from the scalp, and electrooculogram (EOG), which is data regarding changes to corneo-retinal potential between the front and the back of the human eye. 

Convolutional neural network (CNN) is the most successful type of deep-learning model for 2D image analysis such as recognition, classification, and prediction. CNN receives 2D data as an input and extracts high-level features though many hidden convolution layers. Thus, to feed physiological signals into a CNN model, some research works have converted 1D signals into 2D data [4]. Therefore, in this paper we survey 147 contributions which have found highly accurate and significant results of physiological signal analysis using a deep-learning approach. We overview and explain these solutions and contributions in more detail in Section 3, Section 4, and Section 5.

We collected papers via search engine PubMed with keywords combining “deep learning” and a type of physiological signal such as “deep learning electromyogram emg”, “deep learning electrocardiogram ecg”, “deep learning electroencephalogram eeg”, and “deep learning electrooculogram eog” [5]. We found 147 papers published between January 2018 and October 2019 inclusive from various journals and publishers. As illustrated in Figure 1a, the works on EMG, ECG, and EEG using a deep-learning approach have rapidly increased in 2018 and 2019, while EOG and a combination of those signals are limited. Within the works on EEG, there has been an increase by 13, from 33 works in 2018 to 46 works in 2019. As shown in Figure 1b, among the four data modalities of physiological signals, EEG has been conducted in 79 works on a variety of applications. For ECG, there have 47 works conducted. Fifteen works apply to EMG, 1 work to EOG, and 5 works to a combination of those signals.

There are many papers that prove that the deep-learning approach is more successful than traditional machine learning for both implementation and performance. However, this paper does not aim to study the comparison between them. In this paper, we review some recent methods of deep learning in the last two years that analyze the physiological signals. We only compare the key parameters within deep-learning methods such as input data type, deep-learning task, deep-learning model, training architecture, and dataset sources which are involved in predicting the state of hand motion, heart disease, brain disease, emotion, sleep stages, age, and gender.

## 2. Related Works

There are two types of scientific survey in deep-learning approaches regarding physiological signal data for healthcare application between January 2018 and October 2019 inclusive.

The first is oriented to medical fields such as a taxonomy based on medical tasks (i.e., disease detection, computer-aided diagnosis, etc.), or a taxonomy based on anatomy application areas (i.e., brain, eyes, chest, lung, liver, kidney, heart, etc.). Faust et al [6] collected 53 research papers regarding physiological signal analysis using deep-learning methods published from 2008 to 2017 inclusive. This work initially introduced deep-learning models such as auto-encoder, deep belief network, restricted Boltzmann machine, generative adversarial network, and recurrent neural network. Then, it categorized the papers based on types of physiological signal data modalities. Each category points out the medical application, the deep-learning algorithm, the dataset, and the results. 

The second is oriented to deep-learning techniques such as a taxonomy based on deep-learning architectures (i.e., AE, CNN, RNN, DBN, GAN, U-Net, etc.), or the workflow of deep-learning implementation for medical application. Ganapathy et al [3] conducted a taxonomy-based survey on deep learning of 1D biosignal data. This work collected 71 papers from 2010 to 2017 inclusive. Most of the collected papers were published on ECG signals. The goal of the survey was initially to review several techniques for biosignal analysis using deep learning. Then, it classified deep-learning models based on origin, dimension, type of biosignal as an input data, the goal of application, dataset size, type of ground-truth data, and learning schedule of the network. Tobore et al [7], pointed out some biomedical domain considerations in deep-learning intervention for healthcare challenges. It presented the implementation of deep learning in healthcare by categorizing it into biological system, e-health record, medical image, and physiological signals. It ended by introducing research directions for improving health management on a physiological signal application. 

## 3. Physiological Signal Analysis and Modality

Physiological signal analysis is a study estimating the human health condition from a physical phenomenon. There are three types of measurement to record physiological signals: (1) reports; (2) reading; and (3) behavior [8]. The “report” is a response evaluation of questionnaire from subjects who participants in rating their own physiological states. The “reading” is recorded information that is captured by a device to read the human body state such as muscle strength, heartbeat, brain functionality, etc. The “behavior” measurement records a variety of actions such as movement of the eyes. In this paper, we did not review the “report” measurement because the response of “report” is a more biased, less precise question and has broader diversity of question scale. We focus on the technique of “reading” and “behavior” measurement in which the response results are in a signal modality of EMG, ECG, EEG, EOG, or a combination of these signals. 

Table 1 describes the physiological signal modality which was used to implement medical application. The muscle tension pattern of the EMG signal provides hand motion and muscle activity recognition. The variant of heartbeat or heart rhythm provides heart disease, sleep stage, emotion, age, and gender classification. The diversity of brain response of EEG signal provides brain disease, emotion, sleep-stage, motion, gender, words, and age classification. The changes of eye corneo-retinal potential of EOG signal provides sleep-stage classification.

This section presents a categorization of physiological signal data modality regarding the various deep-learning models. We demonstrate key contributions in medical application and performance of systems. We taxonomize contributions such as deep learning on electromyogram (EMG), deep learning on electrocardiogram (ECG), deep learning on electroencephalogram (EEG), deep learning on electrooculogram (EOG), and deep learning on a combination of signals, as shown in Table 2.

We do both a quantitative and qualitive comparison of the deep-learning model. For quantitative comparison, the number of deep-learning models that have been used in medical application is illustrated. For qualitative comparison, since the performance criterion is not provided uniformly, we assume an accuracy value as a base criterion for an overall performance comparison.

### 3.1. Deep Learning with Electromyogram (EMG)

Electromyogram (EMG) signal is data regarding changes of skeleton muscles, which is recorded by putting non-invasive EMG electrodes on the skin such as the commercial MYO Armband (MYB). Since different muscle information is defined by different activity, it can discriminate a pattern of motion such as an open or closed hand. To classify those motion patterns based on the EMG signal information, 15 research works were conducted using deep-learning methods, as shown in Table 3 and Table 4. Within these research works, there are two types of key contribution. One is focused on hand motion recognition and another one is focused on general muscle activity recognition. 

Figure 2 shows the number of deep-learning models used to analyze the EMG signal: (a) illustrates hand motion recognition and (b) illustrates muscle activity recognition. In hand motion recognition, CNN and CNN+RNN models are the most commonly used. In muscle activity recognition, the CNN model is the most commonly used.

Table 3 describes medical application using deep-learning methods in EMG signal analysis from a public dataset source. The publicly available datasets are deployed in the CNN model, which provides overall accuracy >68%. However, the CNN+RNN model provides higher accuracy than the CNN model, with accuracy >82%.

Table 4 describes medical application using deep-learning methods in EMG signal analysis from a private (in-house) dataset source. The works use their own in-house (private) dataset to recognize hand motion. The DBN model performs with overall accuracy >88%. Therefore, The DBN model performs better than CNN and CNN+RNN models. For muscle activity recognition, the CNN model performs NMSE of 0.033 ± 0.017, while RNN/long short-term memory (LSTM) model performs NMSE of 0.096 ± 0.013. Therefore, the CNN model performs better than the RNN/LSTM model.

### 3.2. Deep Learning with Electrocardiogram (ECG)

Electrocardiogram (ECG) is data regarding changes of heartbeat or rhythm. There are 47 research works using deep-learning methods to analyze the ECG signals, as shown in Table 5, Table 6, and Table 7. Their key contributions are categorized as heartbeat signal classification, heart disease classification, sleep-stage classification, emotion detection, and age and gender prediction.

Figure 3 shows the number of deep-learning models used to analyze ECG signal: (a) illustrates heartbeat signal classification in which the CNN model is the most commonly used; (b) illustrates heart disease classification in which CNN is the most commonly used; (c) illustrates sleep-stage detection in which CNN is the most commonly used; (d) illustrates emotion detection in which RNN/LSTM and CNN+RNN are used; and (e) illustrates age and gender classification in which only CNN is used.

Table 5 describes medical application using deep-learning methods in ECG signal analysis from a public dataset source. In heartbeat signal classification, the CNN model performs with overall accuracy >95%. RNN/LSTM model performs with overall accuracy >98%. CNN+RNN/LSTM model performs with overall accuracy >87%. Therefore, RNN/LSTM model performs better than CNN and CNN+RNN/LSTM models. In heart disease classification, CNN model performs with overall accuracy >83%. RNN/LSTM model performs with overall accuracy >90%. CNN+RNN/LSTM model performs with overall accuracy >98%. Therefore, CNN+RNN/LSTM model performs the best. In sleep-stage classification, only CNN model is used and it performs with overall accuracy >87%. 

Table 6 describes medical application using deep-learning methods in ECG signal analysis from a private dataset source. In heartbeat signal classification, only CNN model is used and the CNN model performs with overall accuracy >78%. In heart disease classification, CNN model performs with overall accuracy >97%, while CNN+LSTM model performs with accuracy >83%. Therefore, CNN model performs better than CNN+LSTM model. In sleep-stage classification, CNN and GRU model perform with accuracy of 99%. In emotion classification, CNN+RNN model performs with accuracy >73%. In age and gender prediction, CNN model performs with accuracy >90%.

Table 7 describes medical application using deep-learning methods in ECG signal analysis from a hybrid dataset source. In heartbeat signal classification, CNN+LSTM model performs with accuracy >99%.

### 3.3. Deep Learning with Electroencephalogram (EEG)

Electroencephalogram (EEG) is data regarding changes of the brain measured from the scalp. There are 79 research works using deep-learning methods to analyze the EEG signals, as shown in Table 8, Table 9, and Table 10. Their key contributions are categorized as brain functionality classification, brain disease classification, emotion classification, sleep-stage classification, motion classification, gender classification, word classification, and age classification. 

Figure 4 shows the number of deep-learning models used to analyze EEG signal: (a) illustrates brain functionality classification in which the CNN model is the most commonly used; (b) illustrates brain disease classification in which the CNN is the most commonly used; (c) illustrates emotion classification in which the CNN is the most commonly used; (d) illustrates sleep-stage classification in which CNN is the most commonly used; (e) illustrates motion classification in which CNN is the most commonly used; (f) illustrates gender classification in which only CNN is used; (g) illustrates word recognition in which only AE is used; and (h) illustrates age classification in which only CNN is used.

Table 8 describes medical application using deep-learning methods in EEG signal analysis from a public dataset source. In brain functionality signal classification, CNN model performs with overall accuracy >66%. RNN/LSTM model performs with overall accuracy >77%. CNN+RNN/LSTM model performs with overall accuracy >74%. Therefore, RNN/LSTM model performs better than CNN and CNN+RNN/LSTM models. In brain disease classification, CNN model performs with overall accuracy >93%. RNN/LSTM model performs with overall accuracy >95%. CNN+RNN/LSTM model performs with overall accuracy >90%. Therefore, RNN/LSTM model performs better than CNN and CNN+RNN/LSTM models. In emotion classification, CNN model performs with overall accuracy >55%. RNN/LSTM model performs with overall accuracy >74%. RBM model performs with overall accuracy >75%. Therefore, RBM model performs best. In sleep-stage classification, CNN model performs with overall accuracy >79%. RNN/LSTM model performs with overall accuracy >79%. CNN+RNN/LSTM model performs with overall accuracy >84%. Therefore, CNN+RNN/LSTM model performs better than CNN and RNN/LSTM models. In motion classification, only RNN/LSTM is used, with accuracy >68%. In gender classification, only CNN is used, with accuracy >80%. In word classification, only CNN+AE is used, with overall accuracy >95%.

Table 9 describes medical application using deep-learning methods in EEG signal analysis from a private dataset source. In brain functionality signal classification, CNN model performs with overall accuracy >63%. CNN+RNN/LSTM model performs with overall accuracy >83%. Stacked auto-encoder (SAE)+CNN model performs with overall accuracy >88%. Therefore, SAE+CNN model performs better than CNN and CNN+RNN/LSTM models. In brain disease classification, CNN model performs with overall accuracy >59%. RNN/LSTM model performs with overall accuracy >73%. CNN+RNN/LSTM model performs with overall accuracy >70%. Therefore, RNN/LSTM model performs better than CNN and CNN+RNN/LSTM models. In emotion classification, only CNN+LSTM model is used, with accuracy >98%. In sleep-stage classification, CNN model performs with overall accuracy >95%. CNN+RNN/LSTM model performs with kappa > 0.8. In motion classification, only CNN model is used, with accuracy >80%.

Table 10 describes medical application using deep-learning methods in EEG signal analysis from a hybrid dataset source. In brain disease classification, only the CNN+AE model is used, with kappa > 0.564. In sleep-stage classification, only CNN+LSTM model is used, with kappa > 0.72. In age classification, only the CNN model is used, with accuracy >95%.

### 3.4. Deep Learning with Electrooculogram (EOG)

Electrooculogram (EOG) is data regarding changes of the corneo-retinal potential between the front and the back of the human eye. There are 1 research work using deep-learning methods to analyze the EOG signals, as shown in Table 11. The contribution of deploying deep learning in EOG signal analysis is only for sleep-stage classification. Figure 5 shows the number of deep-learning models which are used to analyze EOG signal for sleep-stage classification. The work used GRU model.

Table 11 describes medical application using deep-learning methods in EOG signal analysis from a public dataset source. In sleep-stage classification, the GRU model performs with accuracy of 69.25%. 

### 3.5. Deep Learning with a Combination of Signals

There are 5 research works using deep-learning methods to analyze a combination of signals, as shown in Table 12. Sokolovsky et al [150] combined EEG and EOG signal. Chambon et al [151] and Andreotti et al [152] combined polysomnography (PSG) signals such as EEG, EMG, and EOG. The work of Yildirim et al [153] exploited the combination signals of EEG and EOG. Croce et al’s [154] contribution was from EEG and magnetoencephalographic (MEG) signals. CNN is used for both sleep-stage classification and the classification of brain and artifactual independent components. Figure 6 shows the number of deep-learning models used to analyze a combination of signals for sleep-stage classification.

Table 12 describes medical application using deep-learning methods in a combination of signals analysis from a public dataset source. In sleep-stage classification, only the CNN model is used, with an overall accuracy >81%.

## 4. Training Architecture

To strive for high accuracy, deep-learning techniques require not only a good algorithm, but also a good dataset [155]. Therefore, the input data is used in two ways: (1) the input data are first extracted as features, then the feature data are fed into the network. Based on our review, some contributions use traditional machine-learning methods as feature extractors described in detail in Section 4.1, while other contributions use deep-learning methods as feature extractors described in detail in Section 4.2; and (2) the raw input data are fed into the network directly for end-to-end learning described in detail in Section 4.3. 

### 4.1. Traditional Machine Learning as Feature Extractor and Deep Learning as Classifier

To distinguish the label of signals, raw signal data is divided into N levels. This step is called feature extraction. Feature extraction is conducted to strengthen the accuracy of prediction in the classification step. Figure 7 illustrates the training architecture using traditional machine learning as feature extractor and deep learning as classifier. For example, the raw EMG signal is divided into N levels using mean absolute value (MAV). The featured data is fed into a CNN to classify hand motion.

Yu et al [9] designed a feature-level fusion to recognize Chinese sign language. The features are extracted using hand-crafted features and learned features from DBN. These two feature levels are concatenated before being fed into the deep belief network and fully connected network for learning. 

For the hand-grasping classification described by Li et al [14], principal component analysis (PCA) method is used for dimension reduction and DNN with a stack of 2-layered auto-encoders, and a SoftMax classifier is applied for classifying levels of force.

Saadatnejad et al [35] proposed ECG heartbeat classification for continuous monitoring. The work extracted raw ECG samples into heartbeat RR interval features and wavelet features. Next, the extracted features were fed into two RNN-based models for classification. 

To classify premature ventricular contraction, Jeon et al [45] extracted the features in the QRS pattern from the ECG signal and classified by modified weight and bias based on the error-backpropagation algorithm. 

Liu et al [60] presented heart disease classification based on ECG signals by deploying symbolic aggregate approximation (SAX) as a feature extraction and LSTM for classification. 

Majidov et al [85] proposed motor imagery EEG classification by deploying Riemannian geometry-based feature extraction and a comparison between convolutional layers and SoftMax layers and convolutional layers, and fully connected layers which outputs 100 units. 

Abbas et al [87] designed a model for multiclass motor imagery classification, in which fast Fourier transform energy map (FFTEM) is used for feature extraction and CNN is used for classification. 

In diagnosing brain disorders, Golmohammadi et al [95] used linear frequency cepstral coefficients (LFCC) for feature extraction and hybrid hidden Markov models and stacked denoising auto-encoder (SDA) model for classifying. 

### 4.2. Deep Learning as Feature Extractor and Traditional Machine Learning as Classifier

Figure 8 illustrates the training architecture of using deep learning as a feature extractor and traditional machine learning as classifier. For example, the raw EEG signal is divided into N levels using SAE. The featured data is fed into support vector machine (SVM) to classify the state of emotion.

Chauhan et al [26] proposed an ECG anomaly class identification algorithm, in which the LSTM and error profile modeling are used as a feature extractor. Then, the multiple choices of traditional machine-learning classifier models were conducted, such as multilayer perception, support vector machine, and logistic regression. 

To diagnose arrhythmia, Yang et al [42] used DL-CCANet and TL-CCANet as feature extractor to discriminate features from dual-lead and three-lead ECGs. Then, the extracted features were fed into the linear support vector machine for classification. 

Nguyen et al [63] proposed an algorithm for detecting sudden cardiac arrest in automated external defibrillators, in which CNN is used as feature extractor (CNNE) and a boosting (BS) classifier. 

Ma et al [131] designed a model to detect driving fatigue. The network model integrated the PCA and deep-learning method called PCANet for feature extraction. Then, SVM/KNN is used for classification.

### 4.3. End-to-End Learning

Rather than extracting the feature from raw data, the raw data is fed into the network for classification. This architecture reduces the feature-extraction step. Figure 9 illustrates the training architecture of using only deep-learning methods to get input raw data, do a classification, and output the result. For example, the ECG data is fed into the LSTM network to classify the states of sudden cardiac arrests.

All works in Table 3, Table 4, Table 5, Table 6, Table 7, Table 8, Table 9, Table 10, Table 11 and Table 12 which are not mentioned in Section 4.1 and Section 4.2 use a raw dataset for end-to-end learning.

## 5. Dataset Sources

We deduce that there are three types of dataset sources used. (1) The public dataset as shown in Table 3, Table 5, Table 8, Table 11, and Table 12 is available online and freely accessible. It has large numbers of samples. Figure 10 illustrates the number of papers using a public dataset based on physiological data modality. For EMG signal analysis, NinaPro DB is the most commonly used. For ECG signal analysis, MIT-BIH is the most commonly used, then PhysioNet is the second most commonly used. For EEG signal analysis, BCI competition II is the most commonly used, then CHB-MIT and DEAP are the second most commonly used. For EOG signal analysis, only PhysioNet is used. For the combination of signal analysis, MASS and PhysioNet is the most commonly used. (2) Private datasets are shown in Table 4, Table 6, and Table 9: it is collected by an author in their own laboratory, hospital, or institution. This dataset requires a specific device for recording or capturing and requires participants or subjects to evolve in the experimental process. Thus, it has a small number of samples. (3) Hybrid datasets are shown in Table 7 and Table 10: the public and private datasets are combined for use in the experiment.

## 6. Discussion

We studied contributions based on types of physiological signal data modality and training architecture. The medical application, deep-learning model, and performance of those contributions have been reviewed and illustrated. 

### 6.1. Discussion of the Deep-Learning Task

In medical application, we deduced that most of the contributions were conducted using a classification task, feature-extraction task, and data compression task. The classification task, which is also known as recognition task, detection task, or prediction task, focuses on whether the instance exists or does not exist. For example, arrhythmia detection [51] analyzes whether the heartbeat signal is normal or arrhythmic. The classification task also focuses on grouping or leveling the types of instances. For example, emotion classification [126] analyzes emotion into groups of sad, happy, neutral, and fear. The feature-extraction task [43] focuses on input data enhancement, in which the unsupervised learning technique is used to label the dataset to avoid a heavy burden from manual labeling. The data compression task [33] focuses on decreasing the data size while still retaining the high quality of data for storage and transmission.

### 6.2. Discussion of the Deep-Learning Model

Even though there are various deep-learning models, we deduced that only CNN, RNN/LSTM, and CNN+RNN/LSTM models are the most commonly used. As theorized in the literature, the RNN/LSTM model predicts continuously sequential data well. However, many contributions convert physiological signals into 2D data and feed those 2D data into a CNN network, in which the performance is good.

### 6.3. Discussion of the Training Architecture

Due to different characteristics of data modality, investigation into the diversity of training architectures has been conducted. The first type of architecture exploits the traditional machine-learning model as a feature extractor and deep-learning model as a classifier. This architecture’s goal is to boost accuracy of classification by converting raw data into feature data. The feature data consists of higher potentially discriminated characteristics than the raw data. The DL classifier trains this feature data in a supervised learning manner. 

In contrast, the second type of architecture employs the deep-learning model as a feature extractor and traditional machine-learning model as a classifier. This architecture’s goal is to reduce the heavy burden of the hand-crafted labeling of the dataset. The DL extractor trains the raw data in an unsupervised learning manner. 

The third architecture type uses only a deep-learning model to train raw data and receive the final output. This architecture’s goal is to not rely on the input dataset, but to strengthen the algorithm of the deep-learning model, in which they believe that the more robust the DL algorithm, the higher the accuracy will be received. This architecture trains raw data in a supervised learning manner. Additionally, this architecture eases the implementation stage. 

In our survey, we could not point out which type of architecture was best. This is because there are no contributions that apply these three types of architecture using the same input dataset for training, testing, and receiving the same desired task.

### 6.4. Discussion of the Dataset Source

We overviewed the sources of the dataset which were conducted for the deep-learning application of physiological signal analysis. The available public datasets which are widely used are MIT-BIH, PhysioNet, BCI competition II, CHB-MIT, DEAP, Bonn University, and NinaPro. The private dataset was collected by authors in their own laboratory, hospital, or institution. The private dataset was collected if the data was not available as a public source. Due to lack of datasets, contributions such as Nodera et al [23] employed a technique of data augmentation, in which a fake dataset is generated by duplicating original data and doing a transformation such as translation and rotation. Contributions [12,16,23,46,58] employed a transfer learning technique. Rather than undertaking a training from a scratch with a huge required dataset, they adapted the pre-weight from a state-of-the-art model such as AlexNet, VGG, ResNet, Inception, or DenseNet.

## 7. Conclusions

In this paper, we conducted an overview of deep-learning approaches and their applications in medical 1D signal analysis over the past two years. We found 147 papers using deep-learning methods in EMG signal analysis, ECG signals analysis, EEG signals analysis, EOG signals analysis, and combinations of signal analysis. 

By reviewing those works, we contribute to the identification of the key parameters used to estimate the state of hand motion, heart disease, brain disease, emotion, sleep stages, age, and gender. Additionally, we reveal that the CNN model predicts the physiological signals at the state-of-the-art level. We have also learned that there is no precise standardized experimental setting. These non-uniform parameters and settings makes it difficult to compare exact performance. However, we compared the overall performance. This comparison should enlighten other researchers to make a decision on which input data type, deep-learning task, deep-learning model, and dataset is suitable for achieving their desired medical application and reaching state-of-the-art level. As a lesson learned from this review, our discussion can also help fellow researchers to make a decision on a deep-learning task, deep-learning model, training architecture, and dataset. Those are the main parameters that effects the system performance.

In conclusion, a deep-learning approach has proved promising for bringing those current contributions to the state-of-the-art level in physiological signal analysis for medical applications.

## Figures and Tables

**Figure 1 sensors-20-00969-f001:**
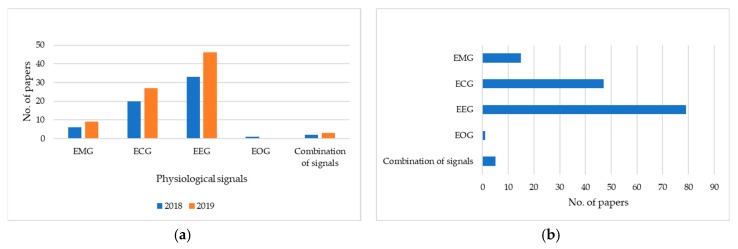
Statistics for papers using a deep-learning approach in physiological signal data grouped by: (**a**) year of publication; and (**b**) data modality.

**Figure 2 sensors-20-00969-f002:**
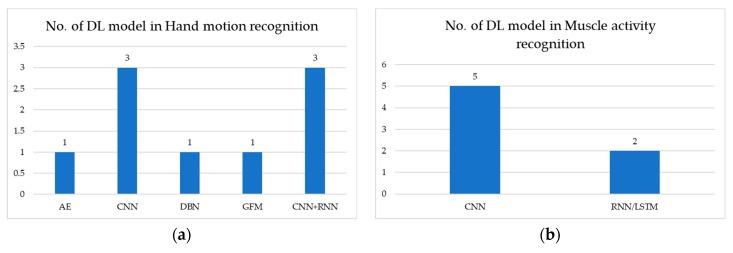
Number of DL models used in EMG signals for: (**a**) hand motion recognition; (**b**) muscle activity recognition.

**Figure 3 sensors-20-00969-f003:**
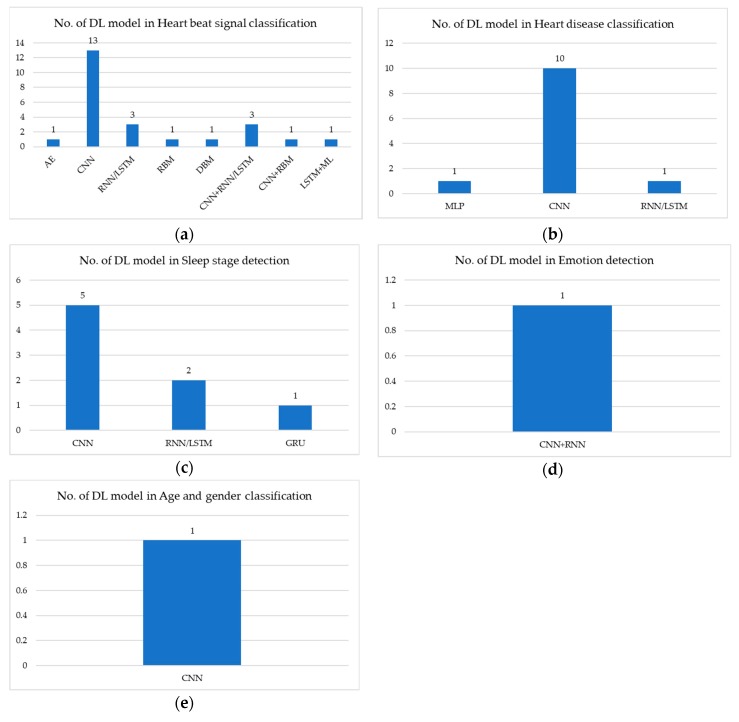
Number of DL models used in ECG signals for: (**a**) heart beat signal classification; (**b**) heart disease classification; (**c**) sleep stage detection; (**d**) emotion detection; and (**e**) age and gender classification.

**Figure 4 sensors-20-00969-f004:**
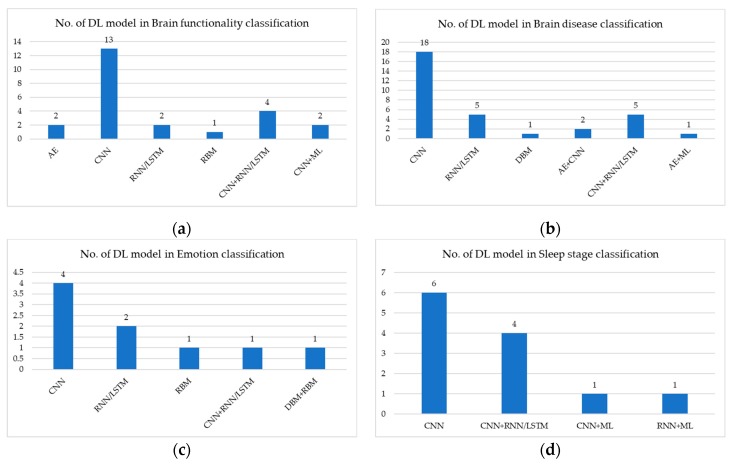
Number of DL models used in EEG signals for: (**a**) brain functionality classification; (**b**) brain disease classification; (**c**) emotion classification; (**d**) sleep-stage classification; (**e**) motion classification; (**f**) gender classification; (**g**) words recognition; and (**h**) age classification.

**Figure 5 sensors-20-00969-f005:**
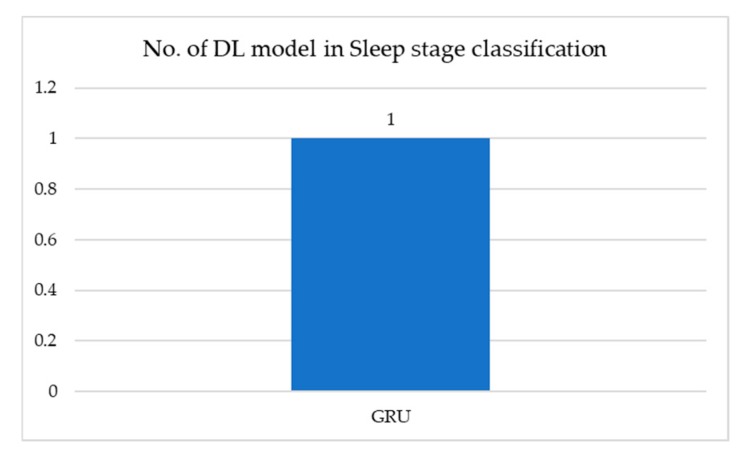
Number of DL models used in EOG signals for sleep-stage classification.

**Figure 6 sensors-20-00969-f006:**
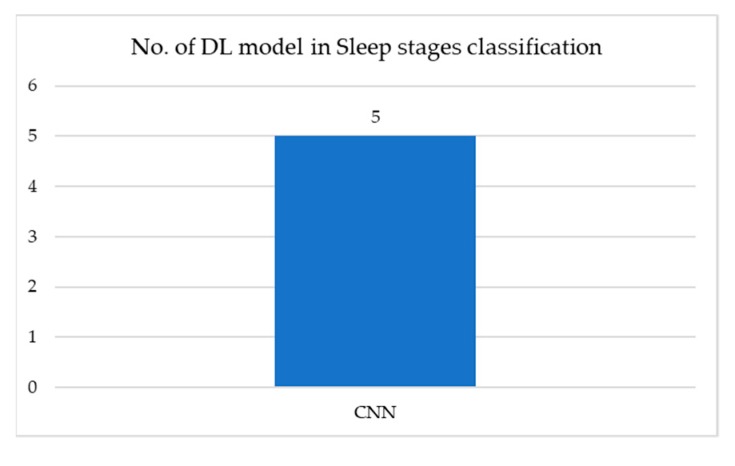
Number of DL models used in a combination of signals for sleep-stage classification.

**Figure 7 sensors-20-00969-f007:**
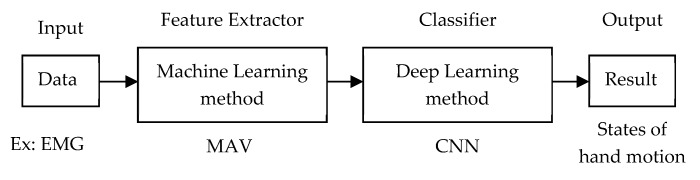
Training architecture of machine learning as feature extractor and deep learning as classifier.

**Figure 8 sensors-20-00969-f008:**
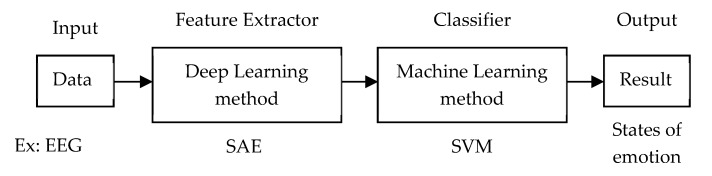
Training architecture of deep learning as feature extractor and machine learning as classifier.

**Figure 9 sensors-20-00969-f009:**
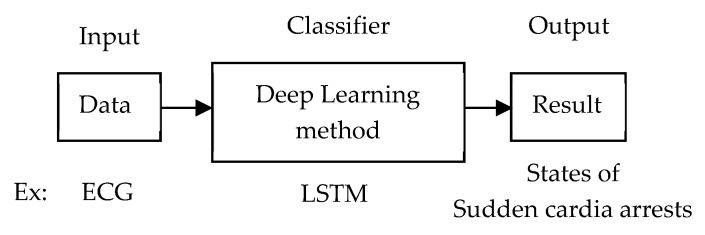
Training architecture of end-to-end learning using deep learning.

**Figure 10 sensors-20-00969-f010:**
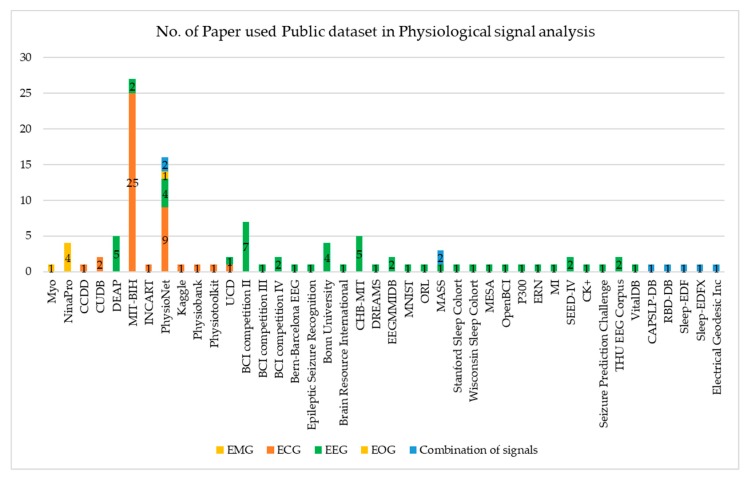
Number of paper used Public dataset in physiological signal analysis.

**Table 1 sensors-20-00969-t001:** Medical application in physiological signal analysis.

Signal Modality	Medical Application
EMG	Hand motion recognition [9,10,11,12,13,14,15,16,17], Muscle activity recognition [18,19,20,21,22,23]
ECG	Heartbeat signal classification [24,25,26,27,28,29,30,31,32,33,34,35,36,37,38,39,40,41,42,43,44,45,46,47,48], Heart disease classification [49,50,51,52,53,54,55,56,57,58,59,60,61,62,63], Sleep-stage classification [64,65,66,67,68], Emotion classification [69], age and gender prediction [70]
EEG	Brain functionality classification [71,72,73,74,75,76,77,78,79,80,81,82,83,84,85,86,87,88,89,90,91], Brain disease classification [92,93,94,95,96,97,98,99,100,101,102,103,104,105,106,107,108,109,110,111,112,113,114,115,116,117,118,119,120,121], Emotion classification [122,123,124,125,126,127,128,129], Sleep-stage classification [130,131,132,133,134,135,136,137,138,139,140,141], Motion classification [142,143,144,145], Gender classification [146], Words classification [147], Age classification [148]
EOG	Sleep-stage classification [149]
Combination of signals	Sleep-stage classification [150,151,152,153,154]

**Table 2 sensors-20-00969-t002:** Table structure based on signal modality and dataset source.

Signal Modality	Public Dataset	Private Dataset	Hybrid Dataset
EMG	Table 3	Table 4	
ECG	Table 5	Table 6	Table 7
EEG	Table 8	Table 9	Table 10
EOG	Table 11		
Combination of signals	Table 12		

**Table 3 sensors-20-00969-t003:** Medical application in EMG analysis using a public dataset source.

Medical Application	Medical Task	DL Model	Dataset Source	No. of Subjects	Performance
Hand motion recognition	Gesture recognition [10]	CNN+RNN	NinaProDB1	27	Accuracy = 87.0%
			NinaProDB2	40	Accuracy = 82.2%
			BioPatRec sub-database	17	Accuracy = 94.1%
			CapgMyo sub-database	18	Accuracy = 99.7%
			csl-hdemg databases	5	Accuracy = 94.5%
	Gesture recognition [11]	CNN	NinaPro	128	Accuracy = 85.78%
			BioPatRec	53	Accuracy = 94.0%
	Gesture signal classification [12]	CNN	MYO	17	Accuracy = 98.31%
			NinaPro	10	Accuracy = 68.98%
	Hand gesture classification [13]	GFM	NinaPro database	10	Accuracy = 63.86 ± 5.12%
	Hand movement classification [16]	CNN+RNN	Ninapro project dataset	78	Accuracy = 87.3 ± 4.9%

**Table 4 sensors-20-00969-t004:** Medical application in EMG analysis using Private dataset source.

Medical Application	Medical Task	DL Model	Dataset Source	No. of Subjects	Performance
Hand motion recognition	Chinese sign language recognition [9]	DBN	6-D inertial sensor (3D-ACC and 3D-GYRO)	8	Accuracy = 95.1% (user-dependent test), Acc = 88.2% (user-independent test)
	Hand-grasping classification [14]	SAE	MYO	15	Accuracy = 95%, SD = 3.58~1.25%
	Hand motion classification [15]	CNN	MYO	7	mean CE ± SD = 9.79 ± 4.57
	Limb movement estimation [17]	CNN+RNN	EMG system (NCC Medical Co., LTD, Shanghai, China)	8	mean R2 = 90.3 ± 4.5%
Muscle activity recognition	Multi-labeled movement information extraction [18]	CNN	ELSCH064NM3 from OT Bioelettronica, Turin, Italy	14	mean exact match rate = 78.7% and a mean Hamming loss = 2.9%
	Muscle activity detection [19]	RNN	Vastus Lateralis and the Lateral Hamstring of a runner	N/A	Signal-to-noise ration < 5
	Musculoskeletal force prediction [20]	CNN	Trigno Wireless EMG system, Delsys, USA	156	RMSE = 0.25, Std. = 0.13
	Prosthetic limb control, Movement Intent decoder [21]	CNN	Grapevine NIP system (Ripple, Salt Lake City, UT, USA)	2	NMSE = 0.033 ± 0.017
		LSTM			NMSE = 0.096 ± 0.013
	Real-time, simultaneous myoelectric control system [22]	CNN	Eight pairs of bipolar surface electrodes (g.HiAmp, g-tec Inc.)	17	Accuracy = 91.61%, Standard error = 0.39
	Wave form identification [23]	CNN	Tokushima University Hospital	83	Accuracy = 86% (test set), Accuracy = 100% (train set)

**Table 5 sensors-20-00969-t005:** Medical application in ECG analysis using a public dataset source.

Medical Application	Medical Task	DL Model	Dataset Source	No. of Subject/Data	Performance
Heartbeat signal classification	Anomaly class identification [26]	LSTM+SVM, LSTM+MLR, LSTM+MLP	MIT-BIH Arrhythmia	43 input features	LSTM+SVM = 42.86% LSTM+MLR = 51.43% LSTM+MLP = 50.0%
	Atrial fibrillation detection [27]	STFT+CNN, SWT+CNN	MIT-BIH Atrial fibrillation	23 annotated ECG recordings	STFT+CNN: Sensitivity = 98.34%, Specificity = 98.24%, Accuracy = 98.29%. SWT+CNN: Sensitivity = 98.79%, Specificity = 97.87%, Accuracy = 98.63%
	CAD ECG signals detection [28]	LSTM+CNN	PhysioNet	47	Accuracy = 99.85%
	Congestive heart failure detection [30]	LSTM	BIDMC-CHF	15	Accuracy = 99.22%
			MIT-BIH NSR	18	Accuracy = 98.85%
			Fantasia	40	Accuracy = 98.92%
	Dofetilide plasma concentrations prediction [31]	CNN	PhysioNet	42	Correlation (r = 0.85)
	ECG Characteristic detection [32]	CNN+RA	QT database (MIT-BIH Arrhythmia+ ST-T Database+ several other ECG databases)	23 records (test set)	P-on = 0.4 ± 14.4 P-peak = −0.4 ± 10.1 P-off = −2.0 ± 12.7 QRS-on = −0.7 ± 10.9 QRS-off = −4.8 ± 13.1 T-peak = −0.3 ± 10.5 T-off = −0.3 ± 18.5
	ECG signal compression [33]	AE	MIT-BIH arrhythmia	48 records	Compression ratio = 106.45, Root mean square difference = 8.00%
	Electrocardiogram diagnosis [34]	CNN+BRNN	Chinese Cardiovascular Disease Database	19K	Accuracy = 87.69%
	Heartbeat classification for continuous monitoring [35]	LSTM	MIT-BIH arrhythmia	N/A	VEB: Accuracy = 99.2%, Sensitivity = 93.0%, Specificity = 99.8% F1 = 95.5% SVEB: Accuracy = 98.3%, Sensitivity = 66.9%, Specificity = 99.8% F1 = 78.8%
	Heartbeat classification [36]	CNN	MIT-BIH Arrhythmia	48 records	Accuracy = 96%, F1-score = 90%
	Heartbeat types classification [37]	CNN+RBM	MIT-BIH arrhythmia	47	AUC = 0.999
	Heartbeats classification [38]	DBLSTM-WS	MIT-BIH arrhythmia	48 records	Accuracy = 99.39%
	Heartbeats classification [39]	CNN	MIT-BIH arrhythmia	48 records	Accuracy = 98.6%
	Multi-lead ECG classification [42]	DL-CCANet, TL-CCANet	MIT-BIH database	48 records	DL-CCANet: Accuracy = 95.2%
			INCART database	78 records	TL-CCANet: Accuracy = 95.52%
	Premature ventricular contraction classification [45]	EBR	MIT-BIH arrhythmia	119 records	Precision = 100%, Recall = 100%, Accuracy = 100%
	Ventricular and supraventricular heart beats detection [47]	RBM+DBM	MIT-BIH database	44 records	Ventricular ectopic beats (Acc = 93.63%), Supraventricular ectopic beats (Acc = 95.57%)
Heart disease classification	Arrhythmia classification [49]	AE+LSTM	MIT-BIH arrhythmia	47	Accuracy = 99.0%, Root mean square difference = 0.70%
	Arrhythmia diagnosis [50]	CNN+LSTM	MIT-BIT arrhythmia	47	Accuracy = 98.10%, Sensitivity = 97.50%, Specificity = 98.70%
	Arrhythmias detection [51]	CNN	MIT-BIH arrhythmia	48	DB1: Accuracy = 97.87% DB2: Accuracy = 99.30%
	Atrial fibrillation (AF) automatically prediction [52]	CNN	MIT-BIH	139 records	Accuracy = 98.7%, Sensitivity = 98.6%, Specificity = 98.7%.
	Beat-wise arrhythmia diagnosis [53]	AE+U-net	MIT-BIH AFDB + PAFDB + MIT-BIH NSRDB	74 (evaluate), 65 (test)	Accuracy = 98.7% Sensitivity = 98.7% Specificity = 98.6%
	Cardiac Arrhythmia classification [54]	MLP, CNN	PhysioBank	208 ECG recordings	Accuracy = 88.7%
			Kaggle		Accuracy = 83.5%
	Cardiac arrhythmias classification [55]	1D-CNN	MIT-BIH Arrhythmia	45	Accuracy = 91.33%
	Cardiologist-Level Arrhythmia detection and classification [56]	CNN	Ziomonitor (iRhythm Technologies Inc, San Francisco, CA)	53,877 patients	AUC = 0.97, Fi-score = 0.837, Sensitivity = 0.780
	Early detection of myocardial ischemia [58]	CNN	PhysioNet	N/A	AUC = 89.6% Sensitivity = 84.4% Specificity = 84.9%, F1-score = 89.2%
	Heart Disease classification [59]	Faster RCNN	MIT-BIH	47	Accuracy = 99.21%
	Heart Diseases classification [60]	LSTM	PhysioNet		Accuracy = 98.4%
	Sudden cardiac arrests (SCA) detection [63]	CNN	Creighton University Ventricular Tachyarrhythmia + MIT-BIH Malignant Ventricular Arrhythmia	35 records + 22 records	Accuracy = 99.26% Sensitivity = 97.07% Specificity = 99.44%
Sleep-stage classification	Apnea detection [64]	CNN	PhysioNet	35	Accuracy = 94.4% Sensitivity = 93.0% Specificity = 94.9%
	Signal quality and sleep position classification [66]	CNN	MIT-BIH arrhythmia	12	C1 class: Precision = 0.99, Recall = 0.99 Sleep position: Precision = 0.99, Recall = 0.99
	Sleep Apnea detection [68]	CNN	PhysioNet Apnea + University College Dublin	70 records + 25 records	Accuracy = 87.6% Sensitivity = 83.1% Specificity = 90.3% AUC = 0.950

**Table 6 sensors-20-00969-t006:** Medical application in ECG analysis using Private dataset source.

Medical Application	Medical Task	DL Model	Dataset Source	No. of Subject/Data	Performance
Heartbeat signal classification	6 types of ECG abnormalities classification [24]	CNN	Telehealth Network of Minas Gerais, Brazil	1,558,415 patients	F1-score > 80% Specificity > 99%
	Cardiologs and veritas detection [29]	CNN	ECGs recorded in the ED of HCMC	1500 records	Cardiologs: Accuracy = 92.2% Sensitivity = 88.7% Specificity = 94.0% Veritas: Accuracy = 87.2% Sensitivity = 92.0% Specificity = 84.7%
	Left ventricular systolic dysfunction detection [41]	CNN	Mayo Clinic ECG	16 056 adult patients	Accuracy = 86.5% Sensitivity = 82.5% Specificity = 86.8%
	Noise detection and screening model [43]	CNN	trauma intensive-care unit	165,142,920 ECG II (10-second lead II electrocardiogram)	Positive prediction = 0.74, Negative prediction = 0.96, Sensitivity = 0.88, Specificity = 0.89, F1-score = 0.80, AUC = 0.93
	Scalogram of ECG classification [46]	ResNet	Physikalisch-Technische Bundesanstalt (PTB)-ECG	290	Accuracy = 0.73
			Chosun University (CU)-ECG	100	Accuracy = 0.94
Heart disease classification	Diabetic subject detection [57]	1D-CNN	Kasturba Medical Hospital (KMH), Manipal, India	30	Accuracy = 97.62%, Sensitivity = 100%
	Heart failure detection on patients in ischemia and post-infarction [61]	CNN	Heart failure database (HFDB)	128 ECG pairs	AUC = 84%
			Ischemia database (IDB)	482 ECG pairs	AUC = 83%
	Mental stress recognition [62]	CNN+LSTM	Zephyr BioHarness 3.0	18	Accuracy = 83.9%, F1-score = 0.81, AUC = 0.92
Sleep-stage classification	Sleep apnea detection [67]	DNN, 1D-CNN, 2D-CNN, RNN, LSTM, GRU	SA dataset	86	Accuracy = 99.0%, Recall = 99.0% (1D-CNN and GRU)
Emotion classification	Stressful state classification [69]	RNN+CNN	Kwangwoon University in Korea	13	Accuracy = 87.39%
			KU Leuven University in Belgium	9	Accuracy = 73.96%
Age and gender prediction	Age and gender prediction [70]	CNN	Mayo Clinic digital data vault	275,056	Accuracy = 90.4%, ACU = 0.97 (independent test data)

**Table 7 sensors-20-00969-t007:** Medical application in ECG analysis using Hybrid dataset source.

Medical Application	Medical Task	DL Model	Dataset Source	No. of Subject/Data	Performance
Heartbeat signal classification	Ventricular fibrillation detection [48]	1D-CNN+ LSTM	PhysioNet MIT-BIH Malignant Ventricular Arrhythmia + Creighton University Ventricular Tachyarrhythmia + American Heart Association ECG Database	N/A	BAC = 99.3%, Sensitivity = 99.7%, Specificity = 98.9%
			OHCA patients	N/A	BAC = 98.0%, Sensitivity = 99.2%, Specificity = 96.7%

**Table 8 sensors-20-00969-t008:** Medical application in EEG analysis using Public dataset source.

Medical Application	Medical Task	DL Model	Dataset Source	No. of Subject/Data	Performance
**Brain functionality classification**	EEG session normal or abnormal detection [74]	1D-CNN+RNN	TUH Abnormal EEG Corpus	1488 abnormal + 1529 normal EEG sessions	Accuracy = 76.9%
	Event-related potential (ERP) detection and analysis [76]	CNN	BCI competition II and III	2	AUC = 0.825 ± 0.064
	Brain activity detection [81]	CNN	BCIC IV 2a. BCI competition IV data set 2a	9	Accuracy = 69%
			BCIC IV 2b. BCI competition IV 2b	9	Accuracy = 83%
			Upper limb movement	15	Accuracy = 31%
	Motor Imagery classification [83]	RNN+3D-CNN	BCI competition IV-2a 4-class Motor Imagery (MI) dataset	9	Accuracy = 74.46%
	Motor Imagery EEG classification [85]	CNN	BCI Competition IV	9	Accuracy = 87.94%
	Motor Imagery EEG Decoding [86]	CP-MixedNet	BCI competition IV 2a	9	Accuracy = 74.6% Precision = 73.9% Recall = 74.7% F1-score = 0.743
			HGD dataset	14	Accuracy = 93.7% Precision = 73.7% Recall = 93.7% F1-score = 0.937
	Multiclass Motor Imagery classification [87]	CNN	BCI Competition Dataset 2a	9	Mean kappa = 0.61 St. Dev = 0.101
	Online decoding of Motor Imagery movement [88]	LSTM, CNN, RCNN	BCI Competition IV	20	LSTM: Accuracy = 66.97 ± 6.45% CNN: Accuracy = 66.2 ± 7.21% RCNN: Accuracy = 77.72 ± 6.5%
	Prediction of bispectral index during target-controlled infusion of propofol and remifentanil [89]	LSTM	vitaldb	180 data points	concordance correlation coefficient (95% CI) = 0.561 (0.560 to 0.562)
	EEG-based BCIs classification [91]	CNN	P300 Evoked Potentials (P300)	8	EEGNet: SNRs = 20.43 DeepCNN: SNRs = 20.50 ShallowCNN: SNRs = 20.53
			Feedback Error-Related Negativity (ERN)	26	EEGNet: SNRs = 20.26 DeepCNN: SNRs = 20.39 ShallowCNN: SNRs = 20.31
			MI	9	EEGNet: SNRs = 25.50 DeepCNN: SNRs = 25.57 ShallowCNN: SNRs = 25.60
**Brain disease classification**	Aberrant epileptic seizure identification [92]	CNN+LSTM	University of Bonn	28	AUC = 0.9703 Accuracy = 90%
	Brain disorders diagnosis [95]	HMM+SDAE	TUH EEG Corpus	13,500 patients	Sensitivity > 90% Specificity < 5%
	Depression screening [97]	CNN	Bonn University	15 normal + 15 depressed patients	Left hemisphere: Accuracy = 93.5% Right hemisphere: Accuracy = 96.0%
	EEG-based epileptic seizure detection [102]	CNN	CHB-MIT dataset	23	Accuracy = 98.3% Sensitivity = 96.7% Specificity = 99.1%
	Epilepsy detection by using scalogram [104]	CNN	Bonn University	A: healthy 100 segment B: healthy 100 segment C: patient 100 segment D: patient 100 segment E: patient 100 segment	A-E: Accuracy = 99.5% A-D: Accuracy = 100% D-E: Accuracy = 98.5% A-D-E: Accuracy = 99.0% A-B-C-D-E: Accuracy = 93.6%
	Epileptic EEG recording classification [106]	CNN	Bern-Barcelona EEG	5	Accuracy = 98.9 ± 0.08%
			Epileptic Seizure Recognition datasets	500	Accuracy = 99.8 ± 0.13%
	Epileptic Seizure prediction [107]	CNN	Seizure Prediction Challenge	5	AUC = 0.79
	Epileptic Seizure prediction [108]	CNN+LSTM	CHB-MIT EEG dataset	22	Accuracy = 99.6%
	Epileptic seizures detection using EEG [110]	LSTM	Bonn University	A: healthy 100 segment B: healthy 100 segment C: patient 100 segment D: patient 100 segment E: patient 100 segment	Accuracy = 100% Sensitivity = 100% Specificity = 100%
	Epileptic seizures prediction [111]	LSTM	Open CHB-MIT Scalp	23	Sensitivity = 100% Specificity = 99.28%
	Seizure detection in multimodal EEG-fNIRs [114]	LSTM	BCI competition IV 2b dataset	40	Sensitivity = 89.7% Specificity = 95.5%
	Seizure Detection [118]	CNN+AE	CHB-MIT dataset	23	Accuracy = 94.37% F1-score = 85.34%
	Seizure detection [119]	LSTM	University of Bonn	A: healthy 100 segment B: healthy 100 segment C: patient 100 segment D: patient 100 segment E: patient 100 segment	Accuracy = 95.54% AUC = 0.9582
**Emotion classification**	Emotion recognition [122]	2D-CNN	DEAP dataset	32	Accuracy = 73.4%
	Emotion Recognition [124]	RNN	SJTU emotion EEG dataset	15	Accuracy = 89.5%
			CK+ facial expression	327 images	Accuracy = 95.4%
	Fear level classification based on emotional dimensions [125]	DNN	DEAP database	32	Accuracy = 59.84% F1-score = 58.78%
	Human emotion recognition [126]	RBM	SEED-IV dataset	15	Accuracy = 85.11%
	Recognition of emotion [127]	DBN-GC+RBM	DEAP dataset	32	Arousal: Accuracy = 75.92% Valence: Accuracy = 76.83%
	Relaxation classification [128]	CNN	OpenBCI	7	1s temporal window: Accuracy = 55.46% 2s temporal window: Accuracy = 98.96%
	Valence and arousal classification [129]	LSTM	DEAP dataset	32	Arousal: Accuracy = 74.65% Valence: Accuracy = 78%
**Sleep-stage classification**	Detect multiple sleep micro-events in EEG [130]	CNN	Montreal Archives of Sleep Studies dataset	19	Precision = 0.3 Recall = 0.95
			Stanford Sleep Cohort dataset	26	Precision = 0.58 Recall = 0.43
			Wisconsin Sleep Cohort dataset	30	Precision = 0.79 Recall = 0.1
			MESA dataset	1000	N/A
	Real-time detection of sleep spindles [133]	CNN+RNN	Montreal archive of sleep studies	19	Sensitivity = 90.07 ± 2.16% Specificity = 96.19 ± 0.71% FDR = 30.36 ± 5.88% F1-score = 0.75 ± 0.05 AUROC = 98.97 ± 0.13%
			DREAMS database	8	Sensitivity = 77.85 ± 4.28% Specificity = 94.2 ± 1.26% FDR = 61.96 ± 7.39% F1-score = 0.48 ± 0.07 AUROC = 95.97 ± 0.96%
	Sleep-stage classification [135]	CNN	PhysioNet (Sleep- EDF dataset)	20	Setting 1: Accuracy = 79.8% Setting 2: Accuracy = 82.6%
	Sleep-stage classification [136]	RNN+SVM	PhysioNet (Sleep-EDF dataset)	20	Setting 1: Accuracy = 79.1% Setting 2: Accuracy = 82.5%
	Sleep-stage classification [137]	CU-CNN	UCD dataset	25	Accuracy = 87% Kappa = 0.8
			MIT-BIH datasets	16 records	Accuracy = 99.9% Kappa = 0.904
	Sleep-stage scoring/detection [138]	CNN+RNN	PhysioNet (Sleep-EDF datasets)	258	Accuracy = 84.26% F1-score = 79.66% Kappa = 0.79
	Sleep stages classification from single-channel EEG [139]	CNN	PhysioNet	8	Accuracy = 98.10%, 96.86%, 93.11%, 92.95%, 93.55%, Kappa = 0.98%, 0.94%, 0.90%, 0.86%,0.89%,
**Motion classification**	Movement intention recognition of disable person [143]	LSTM	MI-based eegmmidb dataset	12	Accuracy = 68.20%
**Gender classification**	Gender prediction from brain rhythms [146]	CNN	Brain Resource International Database	1308	Accuracy > 80% (*p* < 10^−5^)
**Words classification**	Words recognition of speech-impaired people from brain-generated signals [147]	DN-AE-NTM	P300 EEG dataset	9	Accuracy = 97.5%
			EEG recording of individuals with alcoholism and control individuals	64	Accuracy = 95%
			EEGMMIDB	109	Accuracy = 98%
			MNIST	60K samples	Accuracy = 99.4%
			ORL	10 images	Accuracy = 99.1%

**Table 9 sensors-20-00969-t009:** Medical application in EEG analysis using Private dataset source.

Medical Application	Medical Task	DL Model	Dataset Source	No. of Subject/Data	Performance
**Brain functionality classification**	Cerebral Dominance detection [71]	CNN+SVM	Firat University Hospital (Nicolet EEG v32 device)	67	AUC = 0.83 ± 0.05
	Complexity of peri-perceptual processes of familiarity detection [72]	SNN	“Hamrah Clinic” of Tabriz, Iran	20	Accuracy = 83% Sensitivity = 84% Specificity = 86% F1-score = 84%
	Devanagari script input-based P300 speller detection [73]	SAE, DCNN	National Institute of Technology Raipur (ctiCAP Xpress V-amp EEG recorder)	10	Accuracy = 88.22%
	Walking Imagery Evaluation [75]	MMDPN	Biosemi ActiveTwo system	9	Text-MMDPN: AUC = 0.7984 VE-MMDPN: AUC = 0.9424
	EEG event-related classification on children with ADHD from healthy controls [77]	CNN+RNN	Technical University of Dresden	144	Accuracy = 83%
	Focal epileptiform discharges detection [78]	CNN+RNN	Department of Clin. Neurophysiology and Neurology, Medisch Spectrum Twente, Enschede, The Netherlands	50	AUC = 0.94 Sensitivity = 47.4% Specificity = 98.0%
	Human Mental workload Recognition [79]	EL-SDAE	Simulated Human Machine systems	8	Accuracy = 92.02%
	Identify patterns of brain activity of children at idle time and playing videogame time [80]	CNN	University of Houston	233	Accuracy = 67%
	Cross-task mental workload assessment [82]	RNN+3D-CNN	Tsinghua University	20	Accuracy = 88.9%,
	Spectral and temporal feature learning for mental workload assessment [90]	CNN+TCN	Tsinghua University	17	Accuracy = 91.9%,
**Brain disease classification**	Automatic diagnosis of unipolar depression [93]	1D-CNN, 1D-CNN+LSTM	hospital Universiti Sains Malaysia (HUSM)	63	1D-CNN: Accuracy = 98.32% Precision = 99.78% Recall = 98.34% F-score = 97.65% 1D-CNN+LSTM: Accuracy = 95.97% Precision = 99.23% Recall = 93.67% F-score = 95.14%
	Brain disease detection [94]	CNN, RNN, DNN	EEG data of the University of California Irvine	122	CNN: F1-score = 0.94 RNN: F1-score = 0.73 DNN: F1-score = 0.70
	Confusion state induction and detection [96]	CNN	Emotiv Epoc+	16	Accuracy = 71.36%
	Early Alzheimer’s disease diagnosis [98]	DCssCDBM	Beijing Easy monitor Technology	14	Accuracy = 95.04%
	Early prediction of epileptic seizure [99]	CNN+LSTM	Department of Neurology at the First Affiliated Hospital of Xinjiang Medical University	15	Accuracy = 93.40% Sensitivity = 91.88% Specificity = 86.13%
	Early stage Alzheimer disease detection [100]	CNN	Chosun University Hospital (CUH, Gwangju, S. Korea) and Gwangju Optimal Dementia Center located in Gwangju Senior Technology Center (Gwangju, S. Korea)	10	Accuracy = 59.4% Std. = 22.7
	Epileptic discharge detection [105]	CNN	EEG/fMRI study	30	Sensitivity = 84.2%
	Epileptic seizure prediction [109]	CNN	Intracranial electrodes (magenta circles)	10	Sensitivity = 69%
	Identifying Schizophrenia from EEG connectivity Patterns [112]	CNN	Lomonosov Moscow State University	84	Accuracy = 91.69%
	Seizure classification [113]	CNN	Diagnosis of medication refractory TLE based on International League Against Epilepsy (ILAE) criteria	50	Positive Predictio n = 88 ± 7%, Negative Prediction = 79 ± 8%, Accuracy < 50%
	Seizure detection [117]	3D-CNN	Hospital of Xinjiang Medical University	13	Accuracy = 90.00% Sensitivity = 88.90% Specificity = 93.78%
	Seizure detection [120]	CNN	Department of Physiology, College of Medicine, The Catholic University of Korea	249	Sensitivity = 100% Positive Prediction = 98%
	Tracking both the level of consciousness and delirium [121]	CNN+LSTM	Partners Institutional Review Board (IRB)	174	Accuracy = 70% Sensitivity = 69% Specificity = 3% AUC = 0.80
**Emotion classification**	Human Intention Recognition [4]	CNN+LSTM	BCI2000 instrumentation	108 subjects, 3,145,160 EEG records	Accuracy = 98.3%
**Sleep-stage classification**	Driving Fatigue detection from EEG [131]	PCANet+SVM	Guangdong Provincial Work Injury Rehabilitation Center	6	Accuracy = 95%
	Identifying abnormal EEGs, age and sleep-stage classification [132]	CNN	Department of Neurology in Massachusetts General Hospital	8522 EEGs	EEGs: AUC = 0.917 EEGs+Age: AUC = 0.924 EEGs+Age+Sleep: AUC = 0.925
	Sleep stages classification [141]	CNN+LSTM	Chronobiology and Sleep Research, Institute of Pharmacology and Toxicology, University of Zurich, Zurich, Switzerland	75 records	Kappa = 0.8
**Motion classification**	Problem-solving behavioral pattern characterization [144]	CNN	Fakultät Management und Vertrieb, Hochschule Heilbronn Campus Schwäbisch Hall, 74523 Schwäbisch Hall, Germany	26	Accuracy = 99%
	Rapid eye movement behavior disorder [145]	CNN	Center for Advanced Research in Sleep Medicine of the Hôpital du Sacrè- Coeur de Montréal	212	Accuracy = 80 ± 1% AUC = 87 ± 1%

**Table 10 sensors-20-00969-t010:** Medical application in EEG analysis using Hybrid dataset source.

Medical Application	Medical Task	DL Model	Dataset Source	No. of Subject/Data	Performance
**Brain disease classification**	EEG classification of Motor Imagery [101]	CNN + VAE	BCI Competition IV dataset 2b	9	Kappa = 0.564
			Ag-AgCl electrodes	5	3-electrode EEG: Kappa = 0.568 5-electrode EEG: Kappa = 0.603
**Sleep-stage classification**	Real-time sleep-stage classification [134]	CNN+LSTM	SIESTA database	19	Kappa = 0.760 ± 0.022
			Data Science, Philips Research, Eindhoven, Netherlands	29	Kappa = 0.727 ± 0.005
**Age classification**	Age of children classification on performing a verb-generation task, a monosyllable speech-elicitation task [148]	CNN	BCI Competition IV	9	Accuracy = 95%
			University of Toronto, Toronto, Canada	92	

**Table 11 sensors-20-00969-t011:** Medical application in EOG analysis using Public dataset source.

Medical Application	Medical Task	DL Model	Dataset Source	No. of Subject/Data	Performance
Sleep stages classification	Sleep-stage labeling [149]	GRU	PhysioNet	6 sleep stages and 6 sleep disorders	Accuracy = 69.25%

**Table 12 sensors-20-00969-t012:** Medical application in Combine of signals analysis using Public dataset source.

Medical Application	Medical Task	DL Model	Dataset Source	No. of Subject/Data	Performance
Sleep stages classification	Sleep stages classification [150]	CNN	PhysioNet	20	Accuracy = 81% F1-score = 72%
	Sleep-stage classification [151]	CNN	MASS dataset - session 3	62 records	Sensitivity = 85% Specificity = 100%
	Sleep-stage classification [152]	CNN	PhysioNet Sleep-EDF Database (SLPEDF-DB)	19	Kappa = 0.67 ± 0.05
			Montreal Archive of Sleep Studies (MASS-DB)	200	Kappa = 0.74 ± 0.01
			CAP Sleep Database (CAPSLP-DB)	112	Kappa = 0.61 ± 0.01
			RBD Database (RBD-DB)	21	Kappa = 0.48 ± 0.07
	Sleep-stage classification [153]	1D-CNN	Sleep-EDF	9	6 sleep classes: Accuracy = 98.06%, 94.64%, 92.36%, 91.22%, 91.00%
			Sleep-EDFX	61	6 sleep classes: Accuracy = 97.62%, 94.34%, 92.33%, 90.98%, 89.54%
	Classification of brain and artifactual independent component (IC) [154]	CNN	Electrical Geodesic Inc, EEG System Net 300	2048 samples	EEG: Accuracy = 92.4% MEG: Accuracy = 95.4% EEG+MEG: Accuracy = 95.6%

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
