# Peer review of "Deep Learning in Physiological Signal Data: A Survey"

_sensors, 2020, doi:10.3390/s20040969_

Round 1

Reviewer 1 Report

Excellent work. Thank you.

Author Response

Dear reviewer,

I would like to express my sincere thanks for reviewing my paper.

Reviewer 2 Report

I was astonished by this paper, and very rarely saw such a comprehensive review of literature in this domain! I would thank the authors for providing this work to the community, as it will help many other researchers which seek overviews in this domain.   However, to improve the paper, I would suggest the authors to:
- include a section with Skin Conductivity Analysis, as it is a rather wide spread method in physiological signal analysis. I am aware this will require some additional work, and the addition of a section, but the authors would benefit in covering one additional physiological parameter that is typically used in this domain. This increases citations and the wide spreadness of the research work.
- please add the databases that have been searched various literature resources from, as it allows a more objective picture of where research literature is coming from
- please add more diagrammes! e.g. the various texonomies coudl be easily represented as a kind of tree/flow diagramme. Especially the extensive amount of literature reviewed makes it difficult for the reader. Thus please add one of these diagrams for each sections. To state an example: in section 4, the tree could represent the different characterisitcs of deep learning concepts to be utilized with this goal.
- section 4.2. should be a stand alone chapter - e.g. chapter 5, as it reviews various data sources
- however, try to make a reader understand, when to use which methods - this can be done e.g. by integrating the flowcharts as I already described above, or through decision tables or similar
- authors should also include some open source data collection & analysis tools e.g. as https://doi.org/10.1145/2898365.2899801
- please ask yourself - how could you support a researcher in finding the right method to process physiological signals? Add the answer into the paper, then you will get a very high reach with your work. Otherwise you run into the danger, that it will only be one out of many review articles.   I strongly suggest to do another revision of the paper, but for this paper this work will definitely pay off! Try to follow the review comments, and I know it is still a bit of work, but this should not de-courage you. Please see these as constructive to help you in improving the reach of this extremely valuable article. I am looking for the next revision, and your next version. Good luck with this work, it will be a valuable resource for researchers!

Author Response

Dear Professor,

I would like to express my sincere thanks for reviewing my paper.

Reviewer 3 Report

The objectives of this review paper are not sufficiently clear. It seems to focus more on a statistical overview of different concepts w.r.t. the respectively involved bio-signals. A comprehensive CRITICAL & COMPARATIVE ANALYSIS is not sufficiently present in the paper. This pains a lot. 

What is very weak and should be improved is the missing of sufficient and meaningful criteria to be used for a comprehensive performance comparison. In this context, it is clearly not sufficient that ACCURACY is the only performance criteria considered. Other criteria are of interest too, such as: PRECISION, RECALL, F-Measure, etc.
Also, more critical, it is well-known that a "subject-independent" classification endeavour is more challenging and more realistic than a "subject-dependent" one. It would be interesting if the works could consider this aspect in the respective comparisons.

Finally, a comprehensive comparison (qualitative vs quantitative) between "traditional" machine learning approaches and the DL (deep-learning) based ones should be clearly conducted.

Author Response

(The authors gave the same response as above.)
